# Peer review of "Neural Architecture Search with Reinforcement Learning"

_ICLR 2017 — accepted_

[Public Comment · (anonymous) · 30 Nov 2016]
**Request for details on running time and number of machines**

This is a very interesting paper! Can you please provide some information on the amount of time that was required in order to train your model and the hardware that you used?

[Public Comment · (anonymous) · 07 Dec 2016]
**How many networks were trained in total? 102.400 for CIFAR-10?**

This is very interesting work. 
I am wondering how many networks you trained in total. For CIFAR-10, you say you trained the controller for 12,800 iterations. Since every controller used m=8 child replicas in every iteration, does this mean you trained 12.800 * 8 = 102.400 networks in total for CIFAR-10? Thanks!

[Public Comment · (anonymous) · 13 Dec 2016]
**How do you position yourself to: https://arxiv.org/abs/1606.02492 "Convolutional Neural Fabrics"**

They obtain a performance which is worse that reported in your work, however they train their model on a single GPU in contrast to 800 used in the proposed work.

[Official Review · AnonReviewer3 · rating 9 · confidence 5 · 14 Dec 2016]
**A nice paper**

This paper proposed to use RL and RNN to design the architecture of networks for specific tasks. The idea of the paper is quite promising and the experimental results on two datasets show that method is solid.
The pros of the paper are:
1. The idea of using RNN to produce the description of the network and using RL to train the RNN is interesting and promising.
2. The generated architecture looks similar to what human designed, which shows that the human expertise and the generated network architectures are compatible.

The cons of the paper are:
1. The training time of the network is long, even with a lot of computing resources. 
2. The experiments did not provide the generality of the generated architectures. It would be nice to see the performances of the generated architecture on other similar but different datasets, especially the generated sequential models.

Overall, I believe this is a nice paper. But it need more experiments to show its potential advantage over the human designed models.

[Official Review · AnonReviewer4 · rating 9 · confidence 4 · 16 Dec 2016]

This paper presents search for optimal neural-net architectures based on actor-critic framework. The method treats DNN as a variable length sequence, and uses RL to find the target architecture, which acts as an actor. The node selection is an action in the RL context, and evaluation error of the outcome architecture corresponds to reward. A auto-regressive two-layer LSTM is used as a controller and critic. The method is evaluated on two different problems, and each compared with number of other human-created architectures.

This is very exciting paper! Hand selecting architectures is difficult, and it is hard to know how far from optimal results the hand-designed networks are. The presented method is  novel. The authors do an excellent job of describing it in detail, with all the improvements that needed to be done. The tested data represents well the capability of the method. It is very interesting to see the differences between the generated architectures and human generated ones. The paper is written very clearly, and is very accessible. The coverage and contrast with the related literature is done well.

It would be interesting to see the data about the time needed for training, and correlation between time/resources needed to train and the quality of the model. It would also be interesting to see how human bootstrapped models perform and involve.

Overall, an excellent and interesting paper.

[Official Review · AnonReviewer2 · rating 9 · confidence 4 · 17 Dec 2016]

This paper explores an important part of our field, that of automating architecture search. While the technique is currently computationally intensive, this trade-off will likely become better in the near future as technology continues to improve.

The paper covers both standard vision and text tasks and tackle many benchmark datasets, showing there are gains to be made by exploring beyond the standard RNN and CNN search space. While one would always want to see the technique applied to more datasets, this is already far more sufficient to show the technique is not only competitive with human architectural intuition but may even surpass it. This also suggests an approach to tailor the architecture to specific datasets without resulting in hand engineering at each stage.

This is a well written paper on an interesting topic with strong results. I recommend it be accepted.

[Public Comment · (anonymous) · 20 Dec 2016]
**Some relevant related work**

Overall, I find it very exciting that the actor-critic framework can produce such good results. 

There is some related work in Bayesian optimization for architecture search that the authors may not know about. The authors are correct in stating that standard Bayesian optimization methods are limited to a fixed-length space, but there are also some Bayesian optimization methods that support so-called conditional parameters that allow them to effectively search over variable-depth architectures. Two particularly relevant papers are: 

1.

[Public Comment · (anonymous) · 06 Jan 2017]
**Control experiments**

Thank you for the really interesting paper. I have a few questions about the control experiments.

In control experiment 1, how well did the model with the max/sin functions do in terms of perplexity? That seems perhaps more important than how similar the architecture itself is.

What was the distribution over architectures used by random search in control experiment 2? Did you also do grid finetuning of hyperparameters with random search?

[Public Comment · (anonymous) · 06 Jan 2017]
**On validation dataset**

Dear Authors,

Thank you for sharing such an impressive work. I just have a question, in your experimental part, "We then run a small grid search over learning rate, weight decay,
batchnorm epsilon and what epoch to decay the learning rate." May I know that in such a grid search, is the validation accuracy also computed on your 5,000 validation samples from which the reward (used in REINFORCE) is computed?

Thanks a lot!

[Public Comment · (anonymous) · 02 Feb 2017]
**Code available for the design space (not just the final models)?**

Thanks for a very interesting paper!

It is nice to see that the authors will make the code for running the models found by their controller available.
I am wondering whether the code for the design space will also be available. I.e., the spaces from which the policy samples, for CIFAR and Penn Treebank -- e.g. as code with free choices. That would make it much more likely that someone else can reproduce these results. Thanks!

[Public Comment · (anonymous) · 09 Feb 2017]
**Auto-regressive model**

In the paper, you wrote that the controller is auto-regressive and that "each prediction is fed into the next time step as input". What is the input to the controller for the prediction of the first token - a random seed or constant?

Additionally, is the controller an extension of the sequence decoder used in sequence-to-sequence learning? What are the important changes between the NAS controller and the S2S decoder?

Thanks for the excellent paper and congratulations on the acceptance!

[Public Comment · (anonymous) · 25 Apr 2017]
**Inconsistent performance numbers between abstract and section 1 for CIFAR-10 network ?**

The abstract says: "Our CIFAR-10 model achieves a test error rate of 3.65, which is 0.09 percent better and 1.05x faster than
the previous state-of-the-art model" while the last paragraph of section one has: "Our CIFAR-10 model achieves a 3.84 test set error, while being 1.2x faster than the current best model.". Should these be the same numbers or what is the reason for the difference (or is there a newer revision of the paper) ?

[Reviewer Comment · AnonReviewer2 · 19 May 2017 (modified: 02 Jun 2017)]
**Hyperparameters for SotA PTB word level LM**

The paper reports that "[a]fter the controller RNN is done training, we take the best RNN cell according to the lowest validation perplexity and then run a grid search over learning rate, weight initialization, dropout rates and decay epoch. The best cell found was then run with three different configurations and sizes to increase its capacity."

Is it possible for you to include (or provide here) the hyperparameters and type of dropout (i.e. recurrent dropout, embedding dropout, ...) used? Without them, replication would at best require a great deal of trial and error. As with "Recurrent Neural Network Regularization" (Zaremba et al., 2014), releasing a base set of hyper parameters greatly assists in the future work of the field.

This will likely also be desired for the other experiments, such as character LM.

[Final Decision · Program Chairs · 06 Feb 2017]
**ICLR committee final decision**

This was one of the most highly rated papers submitted to the conference. The reviewers all enjoyed the idea and found the experiments rigorous, interesting and compelling. Especially interesting was the empirical result that the model found architectures that outperform those that are used extensively in the literature (i.e. LSTMs).